# Colistin Use in European Livestock: Veterinary Field Data on Trends and Perspectives for Further Reduction

**DOI:** 10.3390/vetsci9110650

**Published:** 2022-11-21

**Authors:** Wiebke Jansen, Jobke van Hout, Jeanine Wiegel, Despoina Iatridou, Ilias Chantziaras, Nancy De Briyne

**Affiliations:** 1Federation of Veterinarians of Europe (FVE), Rue Victor Oudart 7, 1030 Brussels, Belgium; 2Royal GD, Arnsbergstraat 7, 7418 EZ Deventer, The Netherlands; 3Veterinary Epidemiology Unit, Faculty of Veterinary Medicine, Ghent University, Salisburylaan 133, 9820 Merelbeke, Belgium

**Keywords:** colistin, polymyxins, antimicrobial resistance, AMR, antimicrobial use, AMU, animal health, animal welfare

## Abstract

**Simple Summary:**

The antibiotic colistin has been used for Gram-negative bacterial infections in veterinary medicine since the 1950s, with resistance rates remaining low. However, due to the discovery of resistance via transmissible plasmids in 2015, the use of colistin for animals became a subject of discussion, and actions taken resulted in a pan-European reduction of colistin sales for animal use of 76.5% between 2011 and 2020. Nevertheless, few studies evaluated real-world field data on the use patterns of veterinary use of colistin in different countries and species in Europe. A survey among 662 veterinary practitioners revealed that the majority did not use or ceased colistin use (51.9%), 33.4% had decreased their use, 10.4% stabilised their use, and 2.7% increased use. The most important indications for use of colistin were gastrointestinal diseases followed by septicaemia. Governmental/industry restrictions regarding colistin use were reported by 16% of the responding veterinarians, and of them, the most common was “use only after susceptibility testing” (57%).

**Abstract:**

Polymyxin E (colistin) is a medically important active substance both in human and veterinary medicine. Colistin has been used in veterinary medicine since the 1950s. Due to the discovery of the plasmid-borne *mcr* gene in 2015 and the simultaneously increased importance in human medicine as a last-resort antibiotic, the use of colistin for animals was scrutinised. Though veterinary colistin sales dropped by 76.5% between 2011 to 2020, few studies evaluated real-world data on the use patterns of colistin in different European countries and sectors. A survey among veterinarians revealed that 51.9% did not use or ceased colistin, 33.4% decreased their use, 10.4% stabilised their use, and 2.7% increased use. The most important indications for colistin use were gastrointestinal diseases in pigs followed by septicaemia in poultry. A total of 106 (16.0%) responding veterinarians reported governmental/industry restrictions regarding colistin use, most commonly mentioning “use only after susceptibility testing” (57%). In brief, colistin was perceived as an essential last-resort antibiotic in veterinary medicine for *E. coli* infections in pigs and poultry, where there is no alternative legal, safe, and efficacious antimicrobial available. To further reduce the need for colistin, synergistic preventive measures, including improved biosecurity, husbandry, and vaccinations, must be employed.

## 1. Introduction

Polymyxins have been used for over 50 years in both humans and animals [1,2]. Polymyxin B and polymyxin E (colistin) are both medically important active substances [2]. Polymyxins are considered by both the human and animal health sectors as essential. The World Health Organization (WHO) categorises polymyxins as highest priority critically important antimicrobials (HP-CIA) for human medicine [3] and the World Organisation for Animal Health (WOAH) also classifies colistin as veterinary critically important antimicrobials (V-CIA) [4]. In Europe, colistin is authorised for use in animals and has been used in veterinary medicine for decades, especially in pigs, poultry, rabbits, and veal calves to treat infections caused by *Enterobacteriaceae*, mostly given orally as therapeutic group treatment via premix, oral powder, or oral solution but also for individual treatment, i.e., injectable products [5]. Colistin was seen for a long time as a “safe” antibiotic, as it was rarely used in human medicine due to its neuro- and nephrotoxicity and low absorption from the gastrointestinal tract, and the prevalence of resistance to colistin had remained low to very low and was believed to be limited to chromosomal resistance [6,7,8]. However, in 2015, this view changed, and the use of colistin in veterinary medicine is ever since continuously and thoroughly questioned due to the discovery of the plasmid-borne *mcr* gene and its globally distributed variants [9,10]. Simultaneously, colistin became increasingly important in human healthcare and was reclaimed as a last-resort antibiotic to treat certain hospital infections caused by multi-drug resistant (MDR) bacteria, particularly for the highly virulent nosocomial ESKAPE bacteria (*Enterococcus faecium, Staphylococcus aureus*, *Klebsiella pneumoniae, Acinetobacter baumannii, Pseudomonas aeruginosa*, and *Enterobacter* spp.) [11,12]. Consequently, to mitigate the risk of AMR and the risk of transfer of colistin-resistant bacteria to humans, the European legislative framework for colistin use in veterinary medicine was tightened and complemented by a set of additional requirements based on advice or recommendations issued by the European Medicines Agency (EMA) (Table 1). The subsequently following actions resulted in a pan-European reduction of colistin sales by 76.5% (from 10.98 mg/PCU to 2.58 mg/PCU) between 2011 and 2020 though the level of use differs greatly between Member States (MS) [13]. Despite the exhaustive investigation and assessment undertaken by EMA and European Union (EU) MS over the last years as well as by WOAH, investigation and collection of field data on the use of colistin by veterinary practitioners were lacking. To narrow this gap, our survey-based study aimed to analyse data on the concrete use patterns of colistin and investigate trends and developments in use over time, routes of administration, and indications for use in livestock. 

## 2. Materials and Methods

### 2.1. Survey 

A survey was developed on both metaphylaxis (in general) and colistin use in veterinary practice by Federation of Veterinarians of Europe (FVE) together with four European veterinary field experts on poultry, porcine, and bovine health to ensure the content validity of the survey. Questions on colistin were specifically developed together with experts from the Dutch royal animal health service (Gezondheidsdienst voor Dieren, GD). The survey development was reported in the publication by Jerab et al. [21], which summarises the results of the questions covering metaphylaxis. In short, formal testing of the questionnaire was carried out before targeted e-mails were sent to the 51 national FVE member associations requesting to forward the survey to their respective individual members. The survey form was available in nine languages; participation was voluntary and not remunerated. The online survey was accessible between 23 December 2021 and 15 February 2022, with one reminder on 30 January 2022. The complete survey included nine questions, including one multiple-choice question on the administration route of colistin and three questions specifically focusing on colistin use: one open question on the specific indication for which colistin would be chosen as treatment, a multiple-choice question on how their colistin use had changed in time, and a multiple-choice question on whether they were aware of restrictions on the use of colistin as well as an open comment box (Appendix A). The Strengthening the Reporting of Observational Studies in Epidemiology (STROBE) guideline for cross-sectional studies [22] and the Checklist for Reporting Results of Internet E-Surveys (CHERRIES) [23] were used for reporting (Appendix A).

### 2.2. Data Handling and Statistical Analysis

Details on data handling were also reported in detail in the publication by Jerab et al. [21]. In short, data were collected anonymously. Incomplete or duplicate responses based on time stamps were removed. After this validation step, data on terrestrial livestock and avian species were tabulated, processed in Microsoft^®^ Excel, and organised by type of practice (mixed, specialised in pigs, specialised in poultry, specialised in cattle, rabbits, and small ruminants). Comments were thematically categorised for harmonisation purposes. An ordinal logistic regression model was used to evaluate differences between the impact of the independent variables “experience of the veterinarian” (less than 5 years, between 6 and 15 years, between 16 and 25 years, more than 25 years), “type/specialisation of the practice” (mixed practice, practice specialised in poultry, practice specialised in pigs, practice specialised in cattle, practice specialised in small ruminants, practice specialised in cuniculture), and “practice size” (1 to 3 livestock veterinarians, 4 to 6 livestock veterinarians, 7 to 9 livestock veterinarians, more than 10 livestock veterinarians) on the use of colistin (decreased, stable, increased). Calculations were performed in RStudio (package stats, MASS and dbplyr). A *p*-value of ≤0.05 was considered significant.

## 3. Results

A total of 714 responses were received, of which 662 responses met the inclusion criteria (i.e., veterinarians working with terrestrial livestock and/or poultry). 

### 3.1. Demographic Features 

In brief, most responding veterinarians were working in Spain (n = 227, 34.3%), followed by France (n = 108, 16.3%), Germany (n = 61, 9.2%), Hungary (n = 46, 7%), the Netherlands (n = 42, 6.3%), and Poland (n = 35, 5.3%). The majority of responding veterinarians worked in a mixed practice (n = 241, 36.4%), followed by veterinarians working in practices specialised in cattle (n = 166, 25.1%), pigs (n = 141, 21.3%) and poultry (n = 96, 14.5%). Most had ≥25 years of experience (n = 271, 41%), and only a minority of the responding veterinarians had ≤5 years of experience (n = 54, 8.2%), while the number of veterinarians with 6–15 years of experience (n = 178, 24%) was relatively similar to those with 16–25 years of experience (n = 159, 26.9%). The detailed description was reported previously [21].

### 3.2. Trends in Colistin Use

Considering the use of colistin, more than half (n = 344, 51.9%) of the responding veterinarians indicated that they did not use colistin either due to their own choice (I do not use colistin, n = 268, 40.5%) or because colistin use was ceased in their country/sector (n = 76, 11.5%). Of them, less experienced veterinarians (<5 years) were proportionally most likely (n = 31/52, 57.4%) to not use colistin. Ten veterinarians did not respond to this question. Of the remaining respondents who still used colistin, 71.7% (n = 221/308) reduced their use in recent years, 22.4% (n = 69/308) stabilised their use, and 5.8% (n = 18) increased their use. Most of those veterinarians with increased use were working in practices specialised in poultry and mixed practices. An ordinal logistic regression run with three categories (type/specialisation of the practice, size of the practice, and experience of the veterinarian) showed that mixed and poultry practitioners were significantly more likely to still use colistin (p < 0.001, OR 3.94, 95% CI 2.57–6.12 and OR 5.64, 95% CI 3.36–9.55, resp.) (Figure 1). In addition, the model indicated a significant linear increasing trend (*p* = 0.002) to still use colistin across practice sizes (from 1–3 to more than 10 veterinarians working in the same practice). Moreover, those veterinarians with increased use were either very experienced, with more than 25 years of experience (n = 11/18, 61.11%), or not very experienced, with less than 15 years (n = 7/18, 38.9%), but none of the respondents with medium experience increased their use. 

### 3.3. Administration Route of Colistin 

Respondents indicated 254 times their most frequent pattern for metaphylactic treatment that they apply in their practice by ticking the corresponding boxes: colistin and administration route (multiple entries possible per respondent for each species treated). Overall, most veterinarians indicated they used colistin *per os* via drinking water (62.9%, n = 160), with the highest proportion of this administration route in poultry practices at 94.1% (n = 48/51) and the lowest in cattle practices with 34.8% (n = 8/23). Other administration methods used were premixed in feed (n = 39), as top dressing (n = 29), and intramuscular (n = 23). Intramuscular was mentioned only by cattle practitioners and was the second most frequent administration route for cattle (30.4%, n = 7) in our survey (Table 2).

### 3.4. Indications for Colistin Use 

Responding veterinarians were asked for which indications they use, would use, or had used colistin as a treatment of choice, which was responded to for colistin by 440 veterinarians. More than half of them (n = 246, 55.9%) found “no indication” to choose colistin as the treatment of choice. A total of 214 indications for colistin were given (multiple answers possible), of which 69.2% (n = 148) were for gastrointestinal diseases, and of these, diarrhoea (22.9%, n = 49) and colibacillosis (20.1%, n = 43) were the most prominent indications, followed by septicaemia (28.9%, n = 62), with each reported for all livestock species (Figure 2).

### 3.5. Restrictions on the Colistin Use 

A total of 106 responding veterinarians indicated that their decision to use colistin was restricted by governmental/industry irrespective of the country they work. Of these, 74 responses detailed the restrictions, and the most common governmental/industry restriction was found to be that colistin could only be used after an antibiogram and AST (n = 59/106, 53.77%) (Figure 3). 

### 3.6. Individual Comments 

Respondents were also given the opportunity to add free-text comments regarding the use of colistin. Comments were framed in five domains. Most gave more information on their reduction of use (n = 73, 56.2%), and 16 respondents (12.3%) detailed which alternatives were employed to achieve this reduction (mainly vaccination, husbandry, and feed improvements). The imperative for prudent and responsible use (n = 30, 23.1%) was mentioned by illustrating the potential animal health and welfare consequences of a colistin ban (n = 7, 5.4%) and a need for more guidance for the use of colistin (n = 4, 3.1%). Box 1 gives some examples of free-text answers given.

Box 1Examples of free-text comments regarding the use of colistin framed in five domains (reduction of use, alternatives to colistin, responsible use, consequences, and more guidance).
**Reduction of use**
Usage is dramatically decreasing. Extremely sporadic use and only after laboratory dataIt is essential in a variety of indications, but use has been reduced.Laying hens: slightly reduced use, fattening poultry: greatly reduced use of colistinMost colistin use in laying poultry and turkey, but decreasing due to awareness of the need for reduction and due to even more attention for management in all areas
**Alternatives to colistin**
In laying poultry in Spain, the use of colistin has greatly decreased. Alternative available therapies such as vaccines and phytotherapeutics have helped, as well as improvements in other aspects such as nutrition, pro and prebiotics, and management.Overall reduction in the use of oral colistin on neonatal enteritis in calves by increasing the awareness of breeders about the vaccination of mothersVery clear reduction with the introduction of *E. coli* autovaccines in layers
**Responsible use**
I only use colistin in metaphylaxis after an antibiogramAs a last resort, it is still necessaryColistin is essential for managing *Enterobacteriaceae* septicaemia in calves.
**Consequences**
We see no reduction in the use of colistin in the laying hens because we have started keeping the animals under more difficult conditions imposed by market developments (free range chickens, no more beak treatment).Antibiotic treatments in birds are almost non-existent, but when they are carried out, metaphylaxis and colistin are needed
**More guidance**
I prefer a clearer framework for this moleculeIt would be nice if the feed industry also cooperates in stopping colistin that no longer advises

## 4. Discussion

### 4.1. Colistin Use Decreases 

The results of our survey show that practitioners changed prescribing practices, as almost half of the responding veterinarians decreased or ceased their use of colistin in recent years based on the *mcr* gene discovery, the awareness-raising activities on the national and European level, and the EMA guidance [24]. This is supported by the sales data of polymyxins in animals, which declined moderately from 10.98 mg/PCU in 2011 to 9.56 mg/PCU in 2015 (−13%) but drastically as of 2017, with 3.7 mg/PCU (−66%) to 2.58 mg/PCU in 2020, resulting in a total drop of 76.5% between 2011 and 2020 [13]. However, there remains a wide variation among the EU MS in the extent of veterinary use of colistin, with the highest consumption in Cyprus (15.9 mg/PCU), Portugal (11.7 mg/PCU), and Poland (9.1 mg/PCU), whereas Finland, Iceland, and Norway reported no consumption of polymyxins in food-producing animals [13]. These variations were partly explained by differences in animal demographics, availability of alternative antimicrobial agents, dosage regimes, the type of data sources, and veterinarians’ prescribing habits [13]. ==Our results indicated that the larger the practice, the more likely the continued use of colistin. A Dutch report found that porcine veterinarians having more farms as customers as well as those having customers with larger farms had higher antibiotic prescription levels [25]. Farm size has been discussed controversially as a risk factor for higher antimicrobial use (AMU). Some studies showed that larger veal calf and pig farms had a significantly higher treatment frequency with antimicrobials [26,27], while other European studies did not find this association [28,29]. The results also showed higher use by less experienced (merged results from the groups of <5 years and 6–15 years) veterinarians and very experienced veterinarians (>25 years). This is supported by two studies showing that Danish and Dutch less-experienced ruminant veterinarians felt more uncertain in prescribing independently from peers and farmers’ preferences for certain antimicrobials [30,31]. The Danish researchers concluded that the lack of field-generated research of local relevance nourished a culture in which AMU choices are built on personal experience rather than scientific evidence, which also diminished newly educated veterinarians’ self-confidence about their AMU choices [30]. On the other hand, years of experience were associated in the survey on Dutch farm animal veterinarians with being less concerned about the possible contribution of veterinary antibiotic use to antimicrobial resistance, considering it more important to keep the right to prescribe and sell antibiotics and being less hesitant to apply antibiotics to prevent (further dissemination of) animal diseases, which is reflected by the results of our survey [31]. Another study addressing the perceptions of preparedness and acquired skills on key topics related to antimicrobial stewardship (AMS) to final-year veterinary students in Europe came to a similar conclusion. This survey indicated that only 25% of students were familiar with AMU guidelines, and consequently, 75% of the students felt the need for improved teaching on AMS, half of which also demanded more teaching on general antimicrobial therapy [32]. A targeted survey in Serbia and Croatia revealed that only 56.8% of veterinary students were aware of the contribution of AMU in veterinary medicines to overall AMR. In addition, the high importance of some antimicrobials for human medicine was not recognised by surveyed students [33]. This alongside with the results of our survey indicate that the main timeframe of acquiring knowledge and competences on AMU concerning AMR as well as AMS is in the early years of the career. Several possible strategies to improve the quality of veterinary education range from a better link between clinical rotations and the theory taught in pre-clinical modules to a more effective introduction to best practices for AMU as well as continuous education [32,34]. 

To foster evidence-based decision making, many EU countries established national usage and reduction targets, introducing the measurement of prescribing and encouraging AMS. Some countries went further, with others following, starting to collect and also use data, and benchmarking both on the level of veterinary practices and individual farms [35,36]. One of the aims is to reduce the use of HP-CIAs, including colistin. The fact that none of the respondents in our survey with medium experience increased their colistin use is most likely due to the efficacy of the national action plans launched after 2015 as a source of information on AMS for this group. Therefore, the continuous raising of awareness and education of young professionals on AMS is imperative. In addition, the implementation of best practices in husbandry conditions in synergistic combination with alternative options for treatment or prevention, such as vaccinations—including autovaccines—are important elements. 

### 4.2. Indications to Use Colistin

While a great number of respondents did not or no longer use colistin, those that did mainly used colistin for gastrointestinal diseases, with mostly *E. coli* as an etiological cause. Intestinal infections due to Gram-negative bacteria are the main indications for use of colistin in livestock [8,37]. Our survey indicated that mixed practitioners and poultry practitioners were less likely to discontinue colistin use. This is supported by field data on pigs and cattle, including calves, showing that polymyxins were the most-often prescribed antibiotics for diarrhoea with each 40% and 30%, respectively [38]. In particular, colistin acts as the first-choice treatment for neonatal diarrhoea in piglets [39] and veal calves [40] caused by *E. coli*. Colibacillosis was shown to be the major cause of neonatal calf and piglet morbidity and mortality [41,42,43]. The results are therefore consistent with current practice as well as with EMA recommendations to restrict the indications for all veterinary medicinal products containing colistin to be administered orally (in feed or water; calves, sheep, goats, pigs, poultry, rabbits) to “treatment and metaphylaxis of enteric infections caused by susceptible non-invasive *E. coli*.” only [44]. However, our results indicate that some respondents used colistin for infections with *Salmonella* although any indications for other pathogens, prophylaxis, or general indications were removed from the Summary of Product Characteristics and the labels of the authorised veterinary medicines [45]. This might be due to the very limited alternative antimicrobials for certain indications in certain EU countries and depending on susceptibility patterns to other Category B substances depending on resistance profile and disease/patient characteristics. In addition, the European Food Safety Agency (EFSA) assessments of animal diseases caused by bacteria resistant to antimicrobials noted high levels of resistance to first-line antimicrobials (e.g., aminopenicillins, potentiated sulphonamides, tetracyclines) in pathogenic *E. coli* from pigs, poultry, sheep, goats, and calves, suggesting their limited efficacy against these infections in many EU countries [46,47,48,49]. In pigs, high concentrations of in-feed zinc oxide have been extensively used to manage neonatal and weaning gastrointestinal infections [50,51], but this became prohibited in June 2022 due to its negative environmental impact and potential co-selection of resistance genes but might lead to more use again of colistin [52]. In particular, neither are 3rd- and 4th-generation cephalosporins allowed for use in poultry, nor are fluoroquinolones allowed for use in hens laying eggs for human consumption [16,53]. Alongside the withdrawal period of zero days for colistin for eggs [54], this might contribute to why poultry practitioners, especially those working with laying hens, were least likely to cease colistin use for colibacillosis. 

### 4.3. Routes of Administration 

Our survey results showed that oral use (in water, via premixed feed or feed top dressing) is the most used administration method. This is reflected by the number of products licensed in the different EU countries (see EMA Union Product Database [54]), of which most are licensed for oral use. Colistin is not absorbed after oral administration but acts locally on the gut lumen. The initial binding target of polymyxins is the lipopolysaccharide (LPS) in the outer membrane of Gram-negative bacteria, where the “self-promoted uptake” pathway permits the uptake via electrostatic interactions. Thereafter, colistin attaches to the lipid A component of LPS leads to a detergent-like mechanism of action that involves an increase in the permeability of the cell envelope, followed by leakage of periplasmic and cytoplasmic contents, subsequent inner membrane lysis, and ultimately cell death [2]. In addition, colistin binds and neutralises free LPS, conferring a considerable anti-endotoxin activity [55]. However, the supramolecular interaction between colistin and free LPS may even be stronger than between colistin and the intact Gram-negative bacteria with colistin, leading to a competition of binding, which may hamper the antibacterial effects of colistin Gram-negative bacteria but prevent endotoxemia [56]. That explains why *per os* application, particularly in drinking water, is the most common administration route for all species, with the highest proportion in poultry practices and lowest in cattle practices, which reflects the need of group treatment in poultry [21]. Parental use, especially via intramuscular injection, is less frequently used due to the toxic side effects [57] but is still prevalent in some countries and some species, such as ruminants. For cattle practitioners, it was the second most frequent administration route in our survey though the reporting of injectable product sales was limited to a few countries in 2020, including Belgium, France, Italy, Luxembourg, the Netherlands, Romania, and Spain [13]. Intramuscular colistin products seem authorised in Bulgaria, Italy, and Romania, but their use remains a rare exception [54]. 

### 4.4. Restrictions of Use and the Effect of Colistin Reduction in Animals on Human AMR Burden

The continuous effort of veterinary professionals to reduce the need to use colistin in livestock has the potential to lower the burden of AMR in animals concerning the *mcr* prevalence. It was shown that colistin withdrawal as a growth promotor in China resulted in reduced colistin resistance in both animals and humans [58,59]. Whereas growth promotion is banned in the EU, there is evidence that reduced selective pressure can lead to the lower expression of resistance genes and even removal of *mcr* plasmids due to the energy-consuming replication mechanism [7,8,59]. Some governments and sectors already restricted colistin use, for example, the meat poultry sector in the United Kingdom ceased colistin use in 2016 [60], whereas in the EU, use should be based on AST results [18]. However, reliable AST for polymyxins is hampered by the multi-component composition of commercially available polymyxin forms, their poor diffusion in agar due to their large molecular size, their cationic nature, and the development of heteroresistance [61,62]. As a consequence of recent data on polymyxin pharmacokinetics, pharmacodynamics, toxicity and clinical outcomes, AST results for colistin are nowadays mostly reported as minimum inhibitory concentration (MIC) only to emphasise that no MIC is associated with a high probability of treatment success for this antimicrobial [63]. However, exemptions to the need for AST testing before prescription are necessary due to methodological limitations; e.g., if pathogens cannot be cultivated in routine culture systems, no scientifically proven AST method is available for the target pathogen or if sampling would be harmful to the animal (e.g., anaesthesia would be required). Lacking antimicrobial alternatives, colistin was perceived in this survey as a last-resort antibiotic for certain indications. Therefore, current research focuses on structural changes, including complementary compilations to support the treatment or prevention of post-weaning diarrhoea and respiratory infections [64]. The best alternative to antimicrobial treatment (in general) and colistin (in particular) is the prevention of infections, which can be achieved through a synergistic combination of tools such as improving biosecurity and hygiene, and improved husbandry; appropriate nutrition and feed supplements such as probiotics at weaning as well as feed restriction to prevent, e.g., rabbit colibacilloses; and breeding robust animals, healthy parent stock, and regular preventive veterinary visits for monitoring and surveillance of animal health and welfare and to develop herd health plans as well as apply for vaccination programmes as appropriate [65,66,67]. This was also seen in the broader picture of this survey, as described previously [21]. The effect of the use of colistin in animals on human medicine in Europe is estimated to be limited, as chromosomally mediated colistin resistance predominates in human healthcare isolates [68,69,70]. Moreover, the fourteen chromosomal mutations identified bacterial mechanisms mediating colistin resistance were characterised by high stability and irreversibility [14]. It is therefore crucial to continue to only use colistin in both human and veterinary sectors as a precious last-resort antibiotic. In addition, harmonised monitoring on the community level is recommended to provide information on the reservoirs of resistant bacteria that could potentially be transferred to the animals, humans, and the environment interface in all directions and is relevant for both animal and public health [71]. 

### 4.5. Limitations of the Study

The linguistic accessibility and geographical coverage of the survey resulted in sufficient sample size and was supported by multi-language questionnaires. However, even well-translated surveys can be biased by cultural issues. The snowball sampling strategy made it challenging to determine the overall response rate and sampling error or generalise inferences solely based on a purely voluntary call for participation of the obtained questionnaire responses. Furthermore, most of the responses were received from four countries, and six EU countries were without any responses, which affected the representativeness and limits the extrapolation to a full European view. As the use of colistin greatly differs between EU MS, the geographical distribution of responses could influence the results of the survey, leading to a bias. In addition, FVE informed respondents of the survey about its concerns about a wide ban on metaphylaxis, which may result in high morbidity and mortality and devastating production losses. This has the potential to lead to contextual bias, as the survey relied on voluntary responses from practitioners. The request for a follow-up on the results of the survey from 368 of the 662 responding veterinarians represents a large interest from practitioners surrounding the subject of the survey. Despite these limitations, the results of the survey give valuable insight into real-world data on the use of colistin by veterinarians across Europe.

## 5. Conclusions

Colistin remains essential in veterinary medicine and is currently perceived as irreplaceable for specific intestinal enterotoxemic and septic *E. coli* infections in pigs and poultry in countries where there is no alternative legal, safe, and efficacious antimicrobial therapy available. Colistin should be used prudently and responsibly and is currently employed as a last-resort antibiotic uniquely for therapeutic treatments prescribed by a veterinarian following individual AST results to mitigate the risk of AMR whilst acknowledging the limits of AST for colistin. It must be used after physical examination and diagnosis of the animals according to the clinical indication and with respect to species-specific pharmacodynamic and pharmacokinetic properties and the latest scientific evidence. To further reduce the need for colistin in animals and, consequently, the risk of transfer of colistin-resistant bacteria to humans, synergistic preventive measures must be employed, such as improved biosecurity, husbandry, and vaccinations. Harmonised AMR monitoring, including phenotypical and genotypic data sharing between countries as well as between human and veterinary medicine, will be essential for monitoring trends and assessing the attribution of various sources to the molecular epidemiology of colistin resistance as well as the magnitude of transferred AMR bacteria. 

## Figures and Tables

**Figure 1 vetsci-09-00650-f001:**
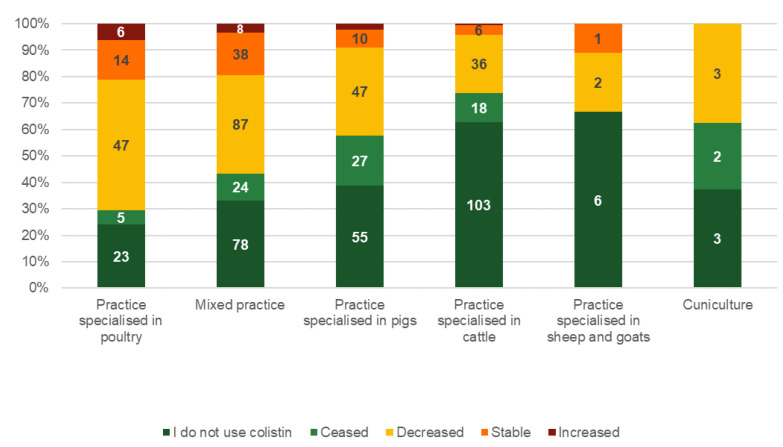
Number (in the bars) and percentages (*y*-axis) of responses of veterinarians on how colistin use changed in recent years per type of practice in descending order of increased colistin use (multiple answers possible if multiple species treated).

**Figure 2 vetsci-09-00650-f002:**
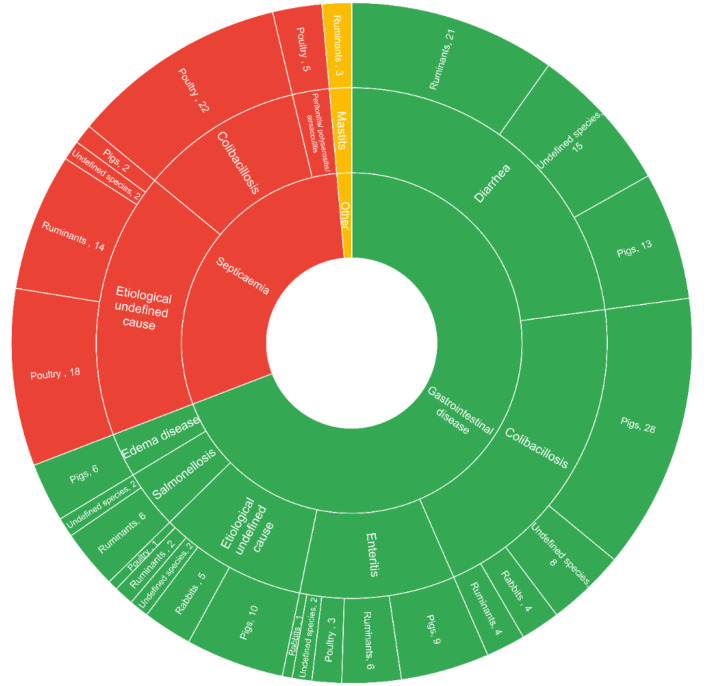
Sunburst chart plotted in Excel showing specific indications (multiple answers possible if multiple species treated) for which colistin was used as a treatment of choice (only livestock species).

**Figure 3 vetsci-09-00650-f003:**
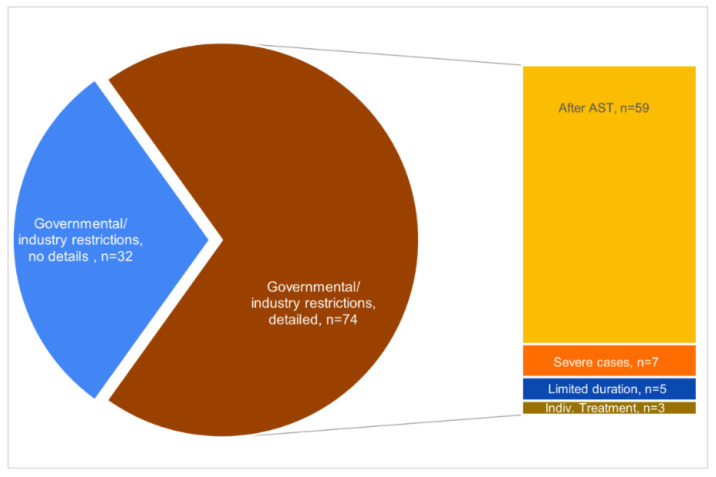
Pie and depending bar chart detailing answers on whether respondents were aware of governmental or industry restrictions and, if so, of which ones.

**Table 1 vetsci-09-00650-t001:** Overview of EU recommendations and legislative decisions on colistin use in veterinary medicine.

Year	Decision/Advice
2006	European-Union-wide ban on antibiotic use for growth promotion, including colistin [14,15].
2013	European Medicines Agency’s advice on colistin [6]: Restriction of the clinical indications for use of colistin to the treatment of enteric infections caused by susceptible non-invasive *E. coli* only; Treatment duration should not exceed 7 days; Any indications for prophylactic use should be removed.
2014	European-Union-wide mandatory antimicrobial susceptibility testing of commensal, indicator, and zoonotic bacteria for colistin isolated from food-producing animals covered by the national monitoring programmes, as laid down in Regulation 2013/652/EU, was implemented.
2016	European Medicines Agency updated advice on colistin following the discovery of the plasmid-borne *mcr*-1 gene in 2015 [16]: Reduce sales to achieve a 65% reduction in European-Union-wide sales of colistin for use in animals; Reduce the use of colistin in animals per country to a maximum of 5 mg colistin/population correction unit (PCU); Reduction should be achieved without an increase in the use of fluoroquinolones, 3rd- and 4th-generation cephalosporins, or overall consumption of antimicrobials.
2016	Based on European Medicines Agency advice, the Commission Implementing Decision laid down the withdrawal of all marketing authorisations for all veterinary medicinal products for oral use containing colistin in combination with other antimicrobial substances [17].
2019	Regulation (EU) 2019/6 laying down authorisation, marketing, and use of medicines in the European Union adopted: Mandatory prescription of all antibiotics in animals; Ban of group prophylactic treatment; Classification of antibiotics and setting additional mandatory requirements before prescription.
2020	Mandatory diagnostic and antimicrobial susceptibility testing before prescription of colistin in animals in the European Union. European Medicines Agency Categorisation of Antimicrobials for animal use classified colistin as Category B “Restrict”, meaning they can only be used if no antibiotics in the lower categories C or D are effective. The use should be based on the results of antimicrobial susceptibility testing whenever possible [18].
2022	European Medicines Agency advice published on the designation of antimicrobials reserved for treatment of certain infections in humans excluded colistin: Due to its necessity to treat serious, life-threatening infections in animals caused by multi-drug resistance Gram-negative bacteria, which inappropriately treated would result in significant morbidity and mortality and impacts on animal welfare; Due to the few alternatives, which are, in most cases, similarly Category B classes. [19]. Based on the European Medicines Agency advice, the European Commission published the corresponding Implementing Act [20].

**Table 2 vetsci-09-00650-t002:** Route of administration for colistin per type of practice.

	Administration Route	Parenteral		*Per os*			Total
Practice Type		Intra-Muscular	Intra- Venous	Drinking Water	Feed Top Dressing	Premixed Feed	
Mixed practice		11	2	62	14	24	113
Practice specialised in cattle		7		8	4	4	23
Practice specialised in pigs		3		40	10	7	60
Practice specialised in poultry		1		48	1	1	51
Practice specialised in sheep and goats		1				2	3
Cuniculture				3		1	4
Grand Total		23	2	160	29	39	254

## Data Availability

The datasets analysed for this study are available on request from the corresponding author, N.D.B. The raw data are not publicly available due to their containing information that could compromise the privacy of research participants.

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
