# Peer review of "Colistin Use in European Livestock: Veterinary Field Data on Trends and Perspectives for Further Reduction"

_vetsci, 2022, doi:10.3390/vetsci9110650_

Round 1
Reviewer 1 Report
This study investigated the use patterns of veterinary use of colistin in Europe. This study is well designed, and the results obtained from this study are reliable. In addition, this study raises alarm bell on the use of antimicrobials in veterinary field.
This study will contribute to public health in human and veterinary medicine, and thus I recommend that this manuscript be accepted for publication in Veterinary Sciences.
However, very minor revisions are required, and I have a few questions.
1. In table 1, the texts should not be centered.
2. In table 2, a diagonal line overlaps “Administration route”, and thus the table should be revised.
3. Colistin is rarely used in human medicine because it has the adverse effect on kidney. On the other hand, the use of colistin is not very rare in veterinary medicine. Isn’t the adverse effect of colistin is not important problem in veterinary medicine?
4. What about the chances that colistin enters the human body via cooked meat? Is colistin heat stable?
Author Response
This study investigated the use patterns of veterinary use of colistin in Europe. This study is well designed, and the results obtained from this study are reliable. In addition, this study raises alarm bell on the use of antimicrobials in veterinary field.
This study will contribute to public health in human and veterinary medicine, and thus I recommend that this manuscript be accepted for publication in Veterinary Sciences.
Thank you very much for your encouraging feedback!
However, very minor revisions are required, and I have a few questions.
- In table 1, the texts should not be centered. Amended.
- In table 2, a diagonal line overlaps “Administration route”, and thus the table should be revised. Amended.
- Colistin is rarely used in human medicine because it has the adverse effect on kidney. On the other hand, the use of colistin is not very rare in veterinary medicine. Isn’t the adverse effect of colistin is not important problem in veterinary medicine?
Thank you for this question! Indeed, the mechanism of colistin nephrotoxicity is via an increase in tubular epithelial cell membrane permeability, which results in cation, anion and water influx leading to cell swelling and cell lysis, applies therefore to all mammals and rodents are often used as animal model for colistin nephrotoxicity1,2. However, we believe that it is assumed that the nephrotoxicity of colistin in human medicine is mainly encountered in intensive care units and is linked to a high intravenous dosage regime with unfavourable predisposing patient-related factors such as age, hypoalbuminemia, hyperbilirubinemia, underlying disease and severity of patient illness as well as co-administration of other nephrotoxic drugs. In contrast, animals are generally treated orally with colistin, and therefore it has less systemic adverse effects as colistin is not absorbed from the intestine. In addition, the animals are in a generally less critical state than the humans treated in the ICUs and the animals are in most cases as well relatively young. We believe that these factors contribute to few reports on nephrotoxicity in livestock.
1Dai, C., Li, J., Tang, S., Li, J., & Xiao, X. (2014). Colistin-Induced Nephrotoxicity in Mice Involves the Mitochondrial, Death Receptor, and Endoplasmic Reticulum Pathways. Antimicrobial Agents and Chemotherapy, 58(7), 4075-4085. https://doi.org/10.1128/AAC.00070-14
2Heybeli, C., Oktan, M. A., & Çavdar, Z. (2019). Rat models of colistin nephrotoxicity: previous experimental researches and future perspectives. European journal of clinical microbiology & infectious diseases : official publication of the European Society of Clinical Microbiology, 38(8), 1387–1393. https://doi.org/10.1007/s10096-019-03546-7
- What about the chances that colistin enters the human body via cooked meat? Is colistin heat stable?
The use of colistin follows strict rules, including so-called mandatory withdrawal periods of 5-7 days before slaughter (depending on the posology of the product) to avoid antibiotic residues in edible tissues. It was shown that oral treatment results in minimal risk of antimicrobial residues in the meat associated with such a short withdrawal timeven in higher concentrations for gastrointestinal infection in laying hens3.
In the EU, both the farmers and their veterinarians charged with monitoring have to supply all relevant information to the official veterinarian at the slaughterhouses on the control of residues as laid down in Chapter III of Council Directive 96/23/EEC. Therefore, edible tissue on the European market, including cooked meat, will not contain any colistin.
3Goetting V, Lee KA, Tell LA. Pharmacokinetics of veterinary drugs in laying hens and residues in eggs: a review of the literature. J Vet Pharmacol Ther. (2011) 34:521–56. doi: 10.1111/j.1365-2885.2011.01287.x
Reviewer 2 Report
This is a study based on a survey on the consumption of colistin in recent years in the European Union. Given the relevance of AMR in our society, the use of colistin in veterinary medicine has been very controversial. The results obtained with this study reflect the responsible consumption of colistin by veterinarians.
The article is very well presented and written, its reading is very enjoyable and the results are described in a very simple but complete way.
I only have a few small recommendations to improve the article:
- L67. Please write Table 1 in full. Modify it also in L140, L169 and L179.
- L70. Please change EU to European Union (EU).
- Table 1. In my opinion, the text tabulated to the right or justified would be better. On the other hand, as a table, the meanings of the acronyms used (EMA, EU, AMEG, MDR and AST) must be included below.
-L80. Please change FVE to Federation of Veterinarians of Europe (FVE). Do the same with GD on line 82, with AMS on L228, VMPs on L265, EFSA on L274 and UK on L321.
-L95. In the text, the meaning of STROBE is indicated in parentheses. Please write it backwards, first the full name and then the acronym in parentheses, just as you have done with CHERRIES.
-L113. Please put the p of p-value in italics. Do the same in L140, L141
-L140. I think the dot behind colistin should be removed.
-L148. I think "colistin use" is misspelled, it's missing an "e".
-L153. Please write per in italics as it is Latin. Do the same in L154, Table2 and L303 (per os).
-Table2. Please remove the italics in the table header. I guess it's a format issue, but "Administration route" comes out displaced. I would recommend reducing the font size by just one point, and right justifying the first column.
-Box. It doesn't have a header...
-L260. Please write the complete word "including"
-L337. There is a double space between colistin and (in particular).
Author Response
This is a study based on a survey on the consumption of colistin in recent years in the European Union. Given the relevance of AMR in our society, the use of colistin in veterinary medicine has been very controversial. The results obtained with this study reflect the responsible consumption of colistin by veterinarians.
The article is very well presented and written, its reading is very enjoyable and the results are described in a very simple but complete way.
Thank you very much for your encouraging feedback!
I only have a few small recommendations to improve the article:
- L67. Please write Table 1 in full. Modify it also inab L140, L169 and L179. Amended.
- L70. Please change EU to European Union (EU). Amended.
- Table 1. In my opinion, the text tabulated to the right or justified would be better. On the other hand, as a table, the meanings of the acronyms used (EMA, EU, AMEG, MDR and AST) must be included below. Amended.
-L80. Please change FVE to Federation of Veterinarians of Europe (FVE). Do the same with GD on line 82, with AMS on L228, VMPs on L265, EFSA on L274 and UK on L321. Amended.
-L95. In the text, the meaning of STROBE is indicated in parentheses. Please write it backwards, first the full name and then the acronym in parentheses, just as you have done with CHERRIES. Amended.
-L113. Please put the p of p-value in italics. Do the same in L140, L141. Amended.
-L140. I think the dot behind colistin should be removed. Amended.
-L148. I think "colistin use" is misspelled, it's missing an "e". Amended.
-L153. Please write per in italics as it is Latin. Do the same in L154, Table2 and L303 (per os). Amended.
-Table2. Please remove the italics in the table header. I guess it's a format issue, but "Administration route" comes out displaced. I would recommend reducing the font size by just one point, and right justifying the first column. Amended.
-Box. It doesn't have a header... Amended.
-L260. Please write the complete word "including" Amended.
-L337. There is a double space between colistin and (in particular). Amended.